# Genomic Analysis of Antibiotic Resistance and Virulence Profiles in *Escherichia coli* Linked to Sternal Bursitis in Chickens: A One Health Perspective

**DOI:** 10.3390/vetsci12070675

**Published:** 2025-07-17

**Authors:** Jessica Ribeiro, Vanessa Silva, Catarina Freitas, Pedro Pinto, Madalena Vieira-Pinto, Rita Batista, Alexandra Nunes, João Paulo Gomes, José Eduardo Pereira, Gilberto Igrejas, Lillian Barros, Sandrina A. Heleno, Filipa S. Reis, Patrícia Poeta

**Affiliations:** 1MicroART—Microbiology and Antibiotic Resistance Team, Department of Veterinary Sciences, University of Trás-os-Montes and Alto Douro, 5000-801 Vila Real, Portugal; jessicalribeiro97@gmail.com (J.R.); vanessasilva@utad.pt (V.S.); catarinairfreitas@gmail.com (C.F.); 2LAQV-REQUIMTE—Associated Laboratory for Green Chemistry, Department of Chemistry, University NOVA of Lisbon, 2829-516 Lisbon, Portugal; 3CIMO—Centro de Investigação de Montanha, La SusTEC, Instituto Politécnico de Bragança, 5300-253 Bragança, Portugal; lillian@ipb.pt (L.B.); sheleno@ipb.pt (S.A.H.); freis@ipb.pt (F.S.R.); 4Functional Genomics and Proteomics Unit, Department of Genetics and Biotechnology, University of Trás-os-Montes and Alto Douro, 5000-801 Vila Real, Portugal; 5Department of Veterinary Sciences, University of Trás-os-Montes and Alto Douro, 5000-801 Vila Real, Portugal; p.pinto.96.pp@gmail.com (P.P.); mmvpinto@utad.pt (M.V.-P.); 6AL4AnimalS—Associate Laboratory for Animal and Veterinary Science, University of Trás-os-Montes and Alto Douro, 5000-801 Vila Real, Portugal; jeduardo@utad.pt; 7CECAV—Veterinary and Animal Research Centre, University of Trás-os-Montes and Alto Douro, 5000-801 Vila Real, Portugal; alexandra.nunes@insa.min-saude.pt (A.N.); j.paulo.gomes@insa.min-saude.pt (J.P.G.); 8Food Microbiology Laboratory, Food and Nutrition Department, INSA, Avenida Padre Cruz, 1649-016 Lisbon, Portugal; rita.batista@insa.min-saude.pt; 9Genomics and Bioinformatics Unit, Department of Infectious Diseases, INSA, 1649-016 Lisbon, Portugal

**Keywords:** *Escherichia coli*, broilers, poultry, sternal bursitis, antimicrobial resistance, genetic lineages

## Abstract

Sternal bursitis, also known as “breast blisters”, is a relatively neglected inflammatory condition in chickens that can impact animal welfare and food quality. Its bacterial causes are poorly studied. In this work, we investigated the role of *Escherichia coli* in sternal bursitis by analyzing 36 isolates collected from affected broiler chickens. We examined their resistance to antibiotics, genetic diversity, and the presence of genes that can promote disease. Our results showed that these *E. coli* strains were genetically diverse, and many carried resistance genes and virulence factors that help bacteria infect animals and survive treatments. Even strains that were not resistant to many antibiotics often possessed many virulence genes. Some of these bacteria also carried traits linked to strains that can cause disease in humans, raising public health concerns. These findings suggest that *E. coli* associated with sternal bursitis may act as hidden reservoirs of dangerous traits and should be included in surveillance programs. Our study underlines the importance of monitoring such infections as part of an integrated approach to animal and human health.

## 1. Introduction

*Escherichia coli* is an adaptable microorganism that resides as a commensal in the gastrointestinal tract of animals and humans. However, certain pathogenic variants are associated with a wide range of infections, affecting the bloodstream, urinary tract, and gastrointestinal systems [1,2,3]. These variants are classified into distinct pathotypes, each linked to specific diseases. The main groups include Intestinal Pathogenic *E*. *coli* (IPEC), which causes gastrointestinal disorders, ranging from mild diarrhea to severe colitis, and Extraintestinal Pathogenic *E*. *coli* (ExPEC), which typically inhabits the gut asymptomatically but can cause severe diseases after migrating to other body sites [4]. Within the ExPEC group, notable variants include Avian Pathogenic *E. coli* (APEC), Neonatal Meningitis-associated *E. coli* (NMEC), and Uropathogenic *E. coli* (UPEC) [5]. Understanding the behavior and pathogenicity of these strains is particularly critical, as the rise of antibiotic resistance among them represents an escalating threat to public and animal health within the One Health Framework [6].

In poultry farming, Avian Pathogenic *Escherichia coli* (APEC) poses a significant threat, contributing to high mortality rates, substantial economic losses, and potential foodborne zoonotic risks [7]. Notably, some APEC strains can cause diseases resembling those associated with human ExPEC [8,9]. Among the conditions affecting poultry, sternal bursitis—commonly referred to as “breast blisters”—is a prominent inflammatory disorder involving the sternal bursa. This cystic structure lies superficially between the two pectoralis major muscles on the sternal keel of chickens and turkeys [10,11].

The term “breast blisters” describes encapsulated swellings of the bursa pre-sternalis, which can contain serous fluid (hygroma) or purulent fluid (bursitis sternalis) and are often surrounded by inflamed tissue. These lesions frequently result from infections caused by *Staphylococcus* spp., coliform bacteria, and *Mycoplasma synoviae* [12]. It is critical to distinguish infectious sternal bursitis from its traumatic counterpart, as the former is more prevalent and has distinct pathophysiological features [10]. Traumatic sternal bursitis, also known as “breast buttons,” is characterized by focal ulcerative dermatitis presenting as a localized, circular skin lesion. This condition involves fluid accumulation without the inflammatory exudate typical of infectious bursitis [12].

Domestic chickens, especially broilers, represent one of the most economically important sources of animal protein worldwide. In Europe, poultry production exceeds 13,200 tons annually, with broiler chickens accounting for most of this output [13]. As such, even subclinical or localized conditions can lead to substantial economic losses [14]. Sternal bursitis is a notable cause of carcass condemnation at slaughter, due to the presence of inflammatory lesions in the breast area, which directly compromise meat quality and marketability [10].

Sternal bursitis is influenced by genetics, housing conditions and management practices, and despite advances in genomic studies unraveling resistance and virulence profiles of *E. coli* isolates from poultry, little is known about the genetic mechanisms specific to isolates associated with sternal bursitis. This includes their virulence factors, antimicrobial resistance traits, and potential for cross-species transmission of resistance genes [12,15,16,17]. This study addresses these gaps by providing a comprehensive genomic characterization of *E. coli* isolates associated with sternal bursitis in chickens. By adopting a One Health perspective, the research aims to enhance the understanding of the resistance and virulence dynamics in poultry farming and their implications for public health and sustainable agricultural practices.

## 2. Methodology

### 2.1. Sample Collection

Between November and December 2021, a total of 40 samples were collected from broiler chickens at a single slaughterhouse in Oliveira de Frades, following the detection of sternal bursitis lesions. The number of samples corresponded to all available carcasses with visible lesions during routine post-mortem inspection. To collect pus from the lesions while minimizing external contamination, aseptic techniques were rigorously followed. The skin surrounding the area of lesion was first disinfected with 70% alcohol to remove surface microorganisms. The lesion site was then carefully opened using sterile instruments, and a sterile swab was used to collect purulent material in a consistent and representative manner. After collection, the swabs were placed in transport medium to preserve the integrity of the samples during transport to the laboratory. During the procedure, strict biosafety and aseptic measures were taken to avoid cross-contamination and ensure the reliability of the samples.

### 2.2. E. coli Isolation

Each swab was introduced into tubes with 5 mL of Brain Heart Infusion (BHI) broth (LiofilChem, Via Scozia, Italy) and then cultured at 37 °C for 24 h. Following that time, the samples were plated onto Chromocult^®^ Coliform agar and Chromocult^®^ Coliform agar (ChromoCult, Fontenay sous Bois, France) supplemented with 2 µg/mL of cefotaxime for the isolation of *E. coli* and cefotaxime-resistant *E. coli* (CTX_R_-*E. coli*). A single colony, which exhibited morphological characteristics indicative of *E. coli*, was collected from each sample and examined using standard biochemistry tests, namely IMViC reactions (indole, methyl red, Voges–Proskauer, and citric acid). The distinct *E. coli* strains were kept at −80 °C for future classification.

### 2.3. Antimicrobial Resistance Phenotype Characterization

The antibiotic susceptibility was determined as described by the Clinical and Laboratory Standards Institute (CLSI) guidelines (2024) [18]. The testing conditions included the disk diffusion method using Mueller–Hinton (MH) agar, with a panel of 20 antibiotics, namely, ampicillin (AMP, 10 μg), amoxicillin–clavulanic acid (AUG, 20 + 10 μg), cefepime (FEP, 30 μg), cefotaxime (CTX, 30 μg), cefoxitin (FOX, 30 μg), ceftazidime (CAZ, 30 μg), aztreonam (ATM, 30 μg), doripenem (DOR, 10 μg), ertapenem (ETP, 10 μg), imipenem (IMI, 10 μg), meropenem (MRP, 10 μg), gentamicin (CN, 10 μg), tobramycin (TOB, 10 μg), amikacin (30 μg), kanamycin (K, 30 μg), streptomycin (S, 10 μg), tetracycline (TET, 30 μg), ciprofloxacin (CIP, 5 μg), trimethoprim-sulfamethoxazole (SXT, 1.25 + 23.75 μg), and chloramphenicol (C, 30 μg).

### 2.4. Whole Genome Sequencing Analysis

Twenty strains (JR25, JR26, JR29, JR30, JR32, JR34, JR35, JR37, JR39, JR42, JR43, JR46, JR48, JR50, JR52, JR53, JR54, JR56, JR58, JR60) were selected for whole genome sequencing (WGS) based on their phenotypic resistance profiles, to capture the greatest possible diversity within the collection and prioritize isolates with multidrug-resistant characteristics. A comprehensive analysis of the whole genomes was carried out using the bioinformatics pipeline INNUca v4.2.3-06 (Instituto Nacional de Saúde Doutor Ricardo Jorge, Lisbon, Portugal). MLST typing was also performed using the INNUca pipeline, which compared the genomic sequences against the Achtman MLST database to identify the sequence types (STs) of the strains. To evaluate the genomes for acquired antibiotic resistance genes and chromosomal point mutations, the ResFinder v4.6.0 (Center for Genomic Epidemiology, Technical University of Denmark, Lyngby, Denmark) and the AMRFinderPlus v4.0.3 (National Center for Biotechnology Information, Bethesda, MD, USA) servers were employed. Virulence factors were identified using MetaVF_Toolkit_VFDB2.0 and VirulenceFinder v2.0 (Center for Genomic Epidemiology, Technical University of Denmark, Lyngby, Denmark). PlasmidFinder v2.1 and SerotypeFinder v2.3 (Center for Genomic Epidemiology, Technical University of Denmark, Lyngby, Denmark) were used with default configurations to ascertain the strain’s plasmid and serotype types.

### 2.5. Genotypic Analysis of Antimicrobial Resistance and Virulence Determinants

Genomic DNA was extracted from the remaining 16 *E*. *coli* strains using the boiling method. In summary, an overnight culture of a single colony was suspended in 1 mL of MilliQ water and boiled for 15 min to destroy the cell walls. Subsequently, the solution was subjected to centrifugation at 12,000 rpm for 2 min, after which the resulting pellet was discarded. The polymerase chain reaction (PCR) mixture included 5 μL of PCR buffer, 1.5 μL of MgCl_2_, 1 μL of 2 mM dNTP, 1 μL of each primer, 0.3 μL of Taq DNA polymerase, and 10 μL of DNA template. The volume of each solution was adjusted with sterile distilled water to obtain a final volume of 50 μL.

The existence of genes encoding resistance to β-lactams was investigated through PCR analysis, encompassing genes such as *ampC*, *blaTEM*, *blaSHV*, *blaCTX-M*, *blaCTX-M*-9, *blaIMP*, *blaVIM*, and *blaOXA*. Furthermore, the existence of genes that confer resistance to non-β-lactams was investigated through PCR, including those encoding resistance to aminoglycosides (*aac(3)-*II, *aac(3)*-IV, *aac(6′)-aph(2″)*, *aadA1*, and *aadA5*), tetracycline (*tetA*, and *tetB*), quinolones (*aac(6′)*-Ib), sulfonamides *(dfrA*, *sul1*, *sul2* and *sul3*), and chloramphenicol (*cmlA* and *floR*).

Additionally, the existence of the *int*1 and *int*2 genes, which encode class 1 and class 2 integrases, was investigated through PCR. Moreover, the *E. coli* strains were subjected to a PCR assay to ascertain the existence of genes encoding some virulence factors, including *fimA* (type 1 fimbriae), *hlyA* (hemolysin), *cnf1* (cytotoxic necrotizing factor), *papC* (P fimbriae), and *aer* (aerobactin iron uptake system). Lastly, the major phylogenetic groups (A, B1, B2, or D) were identified through the detection of tree genes (*chuA*, *yjaA*, and TspE4.C2), as explained by Clermont et al. [19]. The specific primer sequences used in this study are described in Table 1.

## 3. Results and Discussion

### 3.1. Prevalence of E. coli Isolated from Sternal Bursitis in Chickens

Between November and December 2021, a total of forty swabs were collected from chickens presenting with purulent bursitis lesions. From these samples, *E. coli* was isolated in 36 cases (90% of all samples), with 35 isolates recovered on Chromocult^®^ Coliform agar and one isolate obtained on Chromocult^®^ Coliform agar supplemented with 2 µg/mL of cefotaxime, which was identified as a CTX_R_- *E. coli*. *E. coli* has also been implicated as a potential causative or opportunistic pathogen [38]. However, this high isolation rate suggests that *E. coli* may not simply act as a secondary colonizer but could be directly involved in the pathogenesis of sternal bursitis. *E. coli* is a common inhabitant of the avian gut microbiota and is frequently associated with extraintestinal infections, including colibacillosis [39,40]. In the absence of specific studies on the prevalence of *E. coli* in sternal bursitis, we examined its occurrence in other poultry lesions for contextual comparison. Notably, *E. coli* has been frequently isolated from cases of arthritis and osteomyelitis in broilers. A study led in Syria described a 72.9% isolation rate of *E. coli* from joint samples of commercial meat chickens exhibiting lameness, with the highest isolation rate (88.3%) from hock joints [41]. Similarly, in Brazil, *E. coli* was identified in cases of vertebral osteomyelitis and arthritis in broilers, indicating its role in musculoskeletal infections [42]. Pericarditis and perihepatitis are also common manifestations of colibacillosis, with *E. coli* being a primary etiological agent. Despite variations in reported prevalence, *E. coli* is consistently implicated in these conditions [43]. Omphalitis, a naval infection in newborn chicks, is another condition in which *E. coli* is prevalent. A study carried out in Egypt revealed that 87.5% of omphalitis cases in broiler chicks were associated with *E. coli*, highlighting its importance in early infections [44]. These findings underscore the opportunistic nature of *E. coli* in various poultry lesions, suggesting that its presence in sternal bursitis could be part of a broader pattern of extraintestinal infections in avian species.

### 3.2. Phenotypic Profile of the E. coli Isolates

The antimicrobial susceptibility patterns observed in *E. coli* isolates from sternal bursitis lesions revealed resistance to 11 out of the 20 tested antibiotics (Figure 1). The most significant resistance rates were found for ampicillin (41.6%) and tetracycline (30.5%). Although the prevalence of resistance in our study was lower, it aligns with prior reports highlighting widespread resistance to these antibiotics in APEC strains. For instance, a study conducted in Nepal revealed that 99.4% of APEC isolates demonstrated resistance to ampicillin, while 81.1% exhibited resistance to tetracycline [16].

Lower resistance rates were noted for amoxicillin-clavulanic acid, streptomycin (13.8%), ciprofloxacin, trimethoprim-sulfamethoxazole, and chloramphenicol (5.5%), cefotaxime, gentamicin, tobramycin, and amikacin (2.7%). Conversely, Kerek et al. revealed high levels of aminoglycoside resistance (61.2%) in chickens [45]. These contrasting results may reflect differences in antibiotic use policies, farming practices, or regional epidemiological trends. For instance, a wide range of antibiotic medications are registered for use in poultry in many countries, including penicillin, tetracycline, aminoglycosides, and sulfonamides. The resistance levels of *E. coli* originating from broilers in these antibiotic classes are higher than 40% in all the countries considered [46].

Notably, resistance was completely absent against third- and fourth-generation cephalosporins (e.g., cefepime, cefoxitin, ceftazidime), monobactams (aztreonam), carbapenems (doripenem, ertapenem, imipenem, meropenem), and kanamycin. This high level of susceptibility is encouraging and likely reflects the restrictive antimicrobial use policies in food-producing animals under current European Union (EU) legislation. Specifically, Regulation (EU) 2019/6 on veterinary medicinal products, effective since January 2022, alongside Commission Implementing Regulation (EU) 2022/1255, explicitly bans the use of certain critically important antimicrobials for human medicine, including carbapenems, penems, monobactams, and fluoroquinolones, in food-producing species. These regulations are part of a One Health approach to preserve the efficacy of last-resort antibiotics by limiting their veterinary use [47].

These findings highlight the positive impact of antimicrobial stewardship practices and the importance of maintaining such policies to preserve the efficacy of last-resort antimicrobials. In contrast, Manageiro et al. reported reduced susceptibility to third-generation cephalosporins among *E. coli* isolates from broilers in Portugal, associated with the widespread dissemination of CTX-M-type ESBLs. Their findings emphasized the role of CTX-M enzymes in the increase of cephalosporin resistance across human and veterinary surroundings, involving commensal bacteria from animals, humans, and the environment, which may also contribute to the increased use of last-resort antibiotics such as carbapenems and colistin [48]. Nonetheless, the detection of even a single cefotaxime-resistant isolate in this sample set—although minimal—warrants attention, particularly given the global trend of increasing cephalosporin resistance in *E. coli* from food animals.

Multidrug resistance—resistance to three or more antimicrobial classes—was recognized in five isolates (13.9%). One isolate (JR29) showed resistance to five antibiotic classes, suggesting a particularly concerning profile. Two isolates (JR30 and JR48) were resistant to four antibiotic classes, and two others (JR43 and JR58) to three. This rate is lower than reported by Bhattarai et al., who found 91.6% multidrug-resistant isolates [16]. Despite the relatively low multidrug resistance rate in our isolates, the detection of multidrug-resistant strains in a lesion type that is typically underreported in surveillance programs is concerning. These isolates may act as silent reservoirs of resistance genes within poultry flocks and the farm environment. Such reservoirs can contribute to horizontal gene transfer and may compromise therapeutic options in both veterinary and human medicine [49,50].

Moreover, the predominance of resistance to long-established and broadly used antibiotics, such as ampicillin and tetracycline, likely reflects the historical or ongoing usage of these agents as growth promoters or for metaphylaxis in poultry. This pattern is consistent with previous studies reporting high resistance to these antibiotic classes in avian *E. coli* isolates [51,52]. These findings emphasize the importance of including less commonly investigated lesions like sternal bursitis in antimicrobial resistance monitoring schemes. Surveillance efforts that focus solely on classical systemic infections may underestimate the diversity and spread of resistance determinants circulating within poultry production systems.

### 3.3. Whole Genome Sequences and Molecular Characterization of the E. coli Isolates

The twenty isolates examined via WGS exhibited a range of resistance genes, chromosomal mutations, MLST types, O-serotypes, and plasmid replicons (Table 2).

Among the 36 *E. coli* isolates analyzed, 16 (44.4%) carried at a minimum one β-lactamase gene, with *blaTEM*-1B being the most common (43.75%), preceded by *blaTEM*-1A (25%) and *blaTEM*-1C (18.75%). The predominance of *blaTEM* variants reflects the widespread dissemination of these enzymes, which have been recognized as a major determinant of β-lactamic antibiotic resistance [53]. Accordingly, in the state of Ceará, 20% of samples collected from chicken carcasses exhibited the resistance gene *blaTEM*-1B [54]. Considering our research, *blaTEM*-1A has not yet been acknowledged among *E. coli* isolated from broilers. However, *blaTEM*-1C has been detected in one *E. coli* strain from poultry in Denmark [55]. The discovery of additional variants and types, including *blaCTX-M-*1 and *blaOXA* (2.8% each), is particularly concerning due to their association with resistance to third-generation cephalosporins and potential zoonotic transfer [6,48,56]. In some other European regions, the *blaCTX-M-*1 gene has been often linked to ESBL-producing *E. coli* in the broiler production chain [57,58]. Their detection in a sternal bursitis isolate is particularly concerning, as it reveals the silent circulation of ESBL genes in extraintestinal niches not typically targeted by surveillance programs. The *blaOXA* gene was also detected in low prevalence (11%) in a study that included *E. coli* isolated from broiler chickens in Egypt [59].

A broad range of non-β-lactam resistance genes was also identified, including genes conferring resistance to chloramphenicol (*catA1*, *cmlA1*, *cmlA5*, *floR*), aminoglycosides (*aph(6)-Id*, *aph(3″)-Ib*, *aadA1*, *aadA2b*, *aadA5*, *aadA9*, *aadA13*, *sat2*), quinolones (*qnrS1*), tetracyclines (*tetA*), sulfonamides (*sul1*, *sul2*, *sul3*), and trimethoprim (*dfrA1*, *dfrA14*, *dfrA17*). Notably, *tetA* was found in nine isolates, highlighting the role of tetracycline as a historic selective agent. The detection of genes like *cmlA*, *floR*, and *catA1*, despite the prohibition of chloramphenicol use in the European Union, suggests possible environmental sources or cross-resistance [60]. The genes *catA1* and *cmlA* were identified amongst *E. coli* isolates from Korean broiler chickens and from cloacal samples from Saudi Arabia [61,62].

The universal presence of genes such as *acrF* and *mdtM*, which encode efflux pumps intrinsic to the *E. coli* core genome, was noted [63]. While not directly linked to phenotypic resistance, these genes represent latent resistance potential that may be triggered by environmental or therapeutic pressures [64]. Similarly, the presence of *ermD* (2 isolates) and *ermE* (17 isolates)—despite no associated macrolide resistance—may reflect horizontal gene transfer events and warrant further attention [65,66].

Disinfectant resistance genes (*qacE*, *qacL*) and metal resistance genes (e.g., mercury: *merC*, *merP*, *merR*, *merT*; copper: *pco* cluster; silver: *sil* cluster; tellurium: *ter* cluster), as well as genes involved in iron acquisition (*sitABCD*) and stress response (*ariR*, *formA*), were also identified. The presence of these genes—particularly those conferring resistance to heavy metals and biocides—suggests strong environmental selective pressures, likely arising from farming practices such as the use of disinfectants and metal-based supplements [67]. Importantly, many of these genes are found on mobile genetic elements that also harbor antimicrobial resistance genes, facilitating co-selection [68]. The *qacE* and *qacL* genes are associated with decreased susceptibility to quaternary ammonium compounds (QACs), which are widely used disinfectants in poultry farms [69]. Our results demonstrated that 40% of the *E. coli* isolates exhibited at least one heavy metal-resistant gene. Recent studies have indicated an increase in both antibiotic resistance and heavy metal tolerance, suggesting that the bearing of metal-resistant genes may be beneficial for *E. coli* strains [70]. This linkage underscores the risk that frequent disinfection practices may inadvertently co-select for multidrug-resistant bacteria, reinforcing the need for targeted hygiene protocols and prudent biocide usage.

The *glpT_E448K* mutation was detected in almost all (95%) of the sequenced isolates. This mutation has been linked to fosfomycin resistance, as it affects a transporter responsible for antibiotic uptake [71]. Despite the absence of phenotypic testing for fosfomycin, this finding offers a compelling indication of the potential for underlying resistance. A Nigerian study yielded analogous results, with all *E. coli* isolates from chickens exhibiting the same gene [72]. Mutations in the *gyrA* and *parC* genes—both linked to quinolone resistance—were also identified [73]. The *gyrA* p.S83L mutation has been frequently reported in both clinical and veterinary isolates [74,75]. However, in our understanding, this gene has not been recognized among *E. coli* isolated from broilers. Additional mutations were identified in *uhpT*, *nfsA*, and *cyaA*, with varying levels of known clinical relevance.

All isolates were classified as potentially ExPEC, and the detection of 15 different STs suggests significant genetic diversity. This, combined with antimicrobial, heavy metal, and disinfectant resistance genes in several isolates, supports the inclusion of underreported lesions like sternal bursitis in genomic surveillance strategies.

Serotyping revealed considerable diversity, with no dominant profile. However, some detected serotypes (e.g., O5:H11) have been previously associated with human clinical isolates, raising concerns about their zoonotic potential [76]. APEC isolated from broiler breeders with colibacillosis in Mississippi also presented the O88:H7 serotype [77].

A plasmid analysis also revealed high diversity, with *IncF*-type plasmids (particularly *IncFIB* and *IncFIC*) being the most prevalent. These plasmids are often implicated in the horizontal transfer of both resistance and virulence genes and have been strongly associated with ExPEC strains [78,79]. The co-occurrence of multiple replicons in some isolates underscores their genetic plasticity.

A total of 140 virulence-associated genes were screened across the 20 *E. coli* isolates (the full distribution is provided in Appendix A), highlighting their genetic complexity and pathogenic potential. On average, 75 virulence-associated genes were identified per isolate, with JR46 carrying the highest number (*n* = 96) and JR32 the fewest (*n* = 58). These genes were categorized by functional role—colonization, secretion, immune evasion, housekeeping, and true virulence factors. Genes that could not be confidently assigned to any of these categories were considered unclassified (others). The distribution of virulence genes across these functional groups is shown in Figure 2. Housekeeping genes were the most frequently identified, followed by those linked to secretion systems and colonization, indicating a capacity for environmental persistence and host interaction. This analysis highlights the multifactorial nature of virulence in the studied isolates, with a predominance of traits facilitating host colonization and persistence. Genes related to adhesion were nearly ubiquitous among the isolates, including the complete *fim* operon, *ecp* cluster, *ompA*, and *csgA*, suggesting that strong adherence to host tissues might play a crucial role in the establishment and persistence of infection in the sternal bursa, possibly facilitating biofilm formation and chronic inflammation [80]. Additionally, components of the Type VI Secretion System (T6SS) were detected in a subset of isolates, potentially conferring a competitive advantage in polymicrobial environments or playing a role in inter-bacterial interactions during colonization of inflamed tissue [81]. Although often overlooked in APEC-related studies, the T6SS may warrant further functional investigation. All isolates encoded core iron acquisition systems such as enterobactin (*ent*), its associated transporters (*fep*, *fes*), and the virulence regulator *espL1*, indicating a conserved capacity for iron scavenging. In contrast, other systems such as salmochelin (*iro*), aerobactin (*iuc*), yersiniabactin (*ybt*), and heme/hemoglobin uptake systems (*chu*/*shu*) were present in a subset of isolates, suggesting variability in high-affinity iron acquisition strategies that may influence host adaptation or virulence [82]. Several isolates also carried canonical ExPEC/APEC virulence markers such as *iss*, *ompT*, *traT*, and *papC*, supporting their classification as avian pathogenic *E. coli*, while the detection of *tsh* (temperature-sensitive hemagglutinin) in some strains further suggests a potential role in respiratory or systemic infections [83]. Despite the presence of a conserved core set of virulence genes, considerable inter-isolate variability was observed which may reflect distinct evolutionary trajectories or stages of adaptation to the bursal niche, potentially influencing their pathogenicity, persistence, and transmissibility. Overall, the wide range of virulence genes—especially in isolates without significant antimicrobial resistance—highlights the importance of monitoring virulence as well as resistance, particularly in less-studied infection sites like the sternal bursa.

The antibiotic resistance profiles, virulence factors, and phylogenetic groups of the remaining 16 *E. coli* isolates are summarized in Table 3. Phenotypically, only one isolate (JR51; 6.3%) exhibited resistance to ciprofloxacin, and another (JR57; 6.3%) to tetracycline. The *tetB* gene was identified in the tetracycline-resistant isolate, while no acquired resistance genes were detected in the remaining isolates. The low phenotypic resistance observed among these 16 isolates contrasts with the broader resistance patterns identified in the WGS group, possibly reflecting lower selective pressure or less frequent horizontal gene acquisition within this subset. This could be attributed to temporal, environmental, or host-specific factors [84].

All 16 isolates harbored the *fimA* gene, and 10 (62.5%) also carried the *aer* virulence factor. The universal presence of *fimA*, a gene encoding type 1 fimbriae, suggests conserved mechanisms of adhesion, which may contribute to colonization and persistence in the bursal tissue. These findings are in accordance with the one presented by Braga et al. and support the potential pathogenic role of these strains, even in the absence of extensive resistance profiles [42]. Interestingly, no integrons were detected in this group, which might explain the overall lower frequency of acquired resistance genes [85]. Concerning phylogenetic classification, the isolates predominantly belonged to phylogroup B1 (*n* = 12; 75%), preceded by groups A (*n* = 2; 12.5%) and D (*n* = 2; 12.5%), with no detection of phylogroup B2. The predominance of phylogroup B1, often associated with commensal or environmentally adapted *E. coli*, could reflect colonization by strains of lower virulence [86]. However, the presence of key virulence factors and the anatomical localization of infection suggest that even strains from traditionally low-virulence groups may act as opportunistic pathogens under suitable conditions.

Compared to the 20 sequenced isolates, which exhibited higher genetic diversity, resistance gene carriage, and plasmid content, these 16 PCR-characterized strains present a more limited but still relevant pathogenic potential. Their susceptibility profile highlights the complexity of *E. coli* population dynamics in poultry lesions, where both commensal-like and multidrug-resistant strains may coexist. Although often overshadowed by multidrug-resistant strains, susceptible isolates harboring conserved virulence genes may still contribute to disease and serve as reservoirs for future gene acquisition—highlighting the need to monitor all *E. coli* populations, regardless of resistance profile.

## 4. Conclusions

This study provides the first comprehensive genetic characterization of *E. coli* isolates recovered from sternal bursitis lesions in broiler chickens. The isolates exhibited considerable genomic diversity, including multiple sequence types and varied profiles of antimicrobial resistance, virulence, and plasmid content. Notably, several isolates carried classical ExPEC/APEC virulence markers and iron acquisition systems, despite lacking extensive antimicrobial resistance, underscoring their potential pathogenicity. The presence of chromosomal mutations and mobile genetic elements further highlights the evolutionary adaptability of these strains. Altogether, our findings suggest that sternal bursitis, though frequently underestimated, may represent a relevant niche for *E. coli* with zoonotic potential. These results reinforce the need to include atypical lesions in surveillance programs and support a broader, One Health-based approach to monitor antimicrobial resistance and virulence in animal pathogens.

## Figures and Tables

**Figure 1 vetsci-12-00675-f001:**
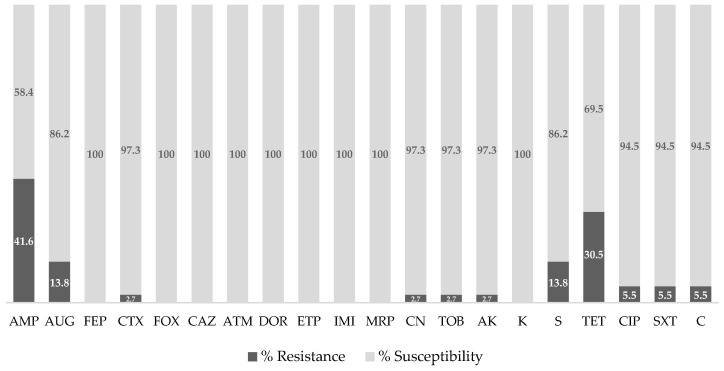
Antibiotic resistance profile of *E*. *coli* isolates from cases of sternal bursitis in broilers (AMP—ampicillin; AUG—amoxicillin-clavulanic acid; FEP—cefepime; CTX—cefotaxime; FOX—cefoxitin; CAZ—ceftazidime; ATM—aztreonam; DOR—doripenem; ETP—ertapenem; IMI—imipenem; MRP—meropenem; CN—gentamicin; TOB—tobramycin; AK—amikacin; K–kanamycin; S—streptomycin; TET—tetracycline; CIP—ciprofloxacin; SXT—trimethoprim-sulfamethoxazole; C—chloramphenicol).

**Figure 2 vetsci-12-00675-f002:**
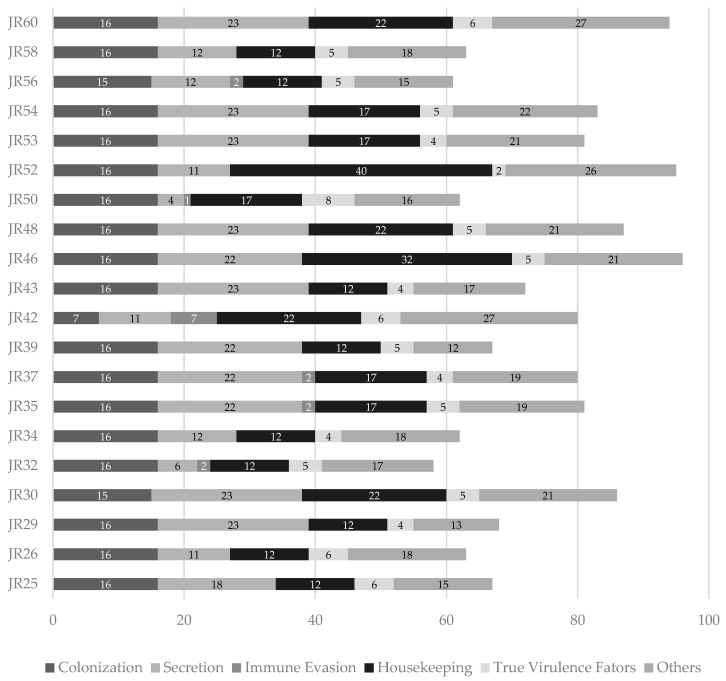
Allocation of virulence-associated genes by functional grouping across the 20 *E. coli* isolates. Genes were grouped into five categories based on predicted biological roles: colonization, secretion, immune evasion, housekeeping, and true virulence factors. Unclassified genes represent those for which no specific function could be determined.

**Table 1 vetsci-12-00675-t001:** Primer sequences used for PCR amplification of antimicrobial resistance genes, virulence factors, integrase genes, and phylogenetic markers in *E. coli* isolates. The table includes the target gene, nucleotide sequence, annealing temperature, and expected amplicon size.

	Target Gene	Primer (5′ → 3′)	Annealing Temperature	Amplicon Size (bp)	Reference
Antibiotic Resistance Genes	*ampC*	AATGGGTTTTCTACGGTCTG	57 °C	191	[20]
		GGGCAGCAAATGTGGAGCAA			
	*blaTEM*	ATTCTTGAAGACGAAAGGGC	60 °C	1150	[21]
		ACGCTCAGTGGAACGAAAAC			
	*blaSHV*	CACTCAAGGATGTATTGTG	50 °C	885	[22]
		TTAGCGTTGCCAGTGCTCG			
	*blaCTX-M*	CGATGTGCAGTACCAGTAA	52 °C	585	[23]
		TTAGTGACCAGAATCAGCGG			
	*blaCTX-M9*	GTGACAAAGAGAGTGCAACGG	62 °C	857	[24]
		ATGATTCTCGCCGCTGAAGCC			
	*blaVIM*	TTTGGTCGCATATCGCAACG	66 °C	500	[25]
		CCATTCAGCCAGATCGGCAT			
	*blaIMP*	GTTTATGTTCATACTCG	45 °C	432	
		GGTTTAAAAAACAACCAC			
	*blaOXA*	CCAAAGACGTGGATG	61 °C	817	[26]
		GTTAAATTCGACCCCAAGTT			
	*aac(3′)-II*	ACTGTGATGGGATACGCGTC	60 °C	237	[27]
		CTCCGTCAGCGTTTCAGCTA			
	*aac(3′)-IV*	CTTCAGGATGGCAAGTTGGT		286	
		TACTCTCGTTCTCCGCTCAT			
	*aac(6′)-aph(2)*	CCAAGAGCAATAAGGGCATA	60 °C	220	
		CACTATCATAACCACTACCG			
	*aadA1*	GCAGCGCAATGACATTCTTG	60 °C	282	[28]
		ATCCTTCGGCGCGATTTTG			
	*aadA5*	CTTCAGTTCGGTGAGTGGC	55 °C	453	[29]
		CAATCGTTGCTTTGGCATAT			
	*tetA*	GTAATTCTGAGCACTGTCGC	62 °C	937	[30]
		CTGCCTGGACAACATTGCTT			
	*tetB*	CTCAGTATTCCAAGCCTTTG	57 °C	416	
		CTAAGCACTTGTCTCCTGTT			
	*aac(6′)-Ib*	TTGCGATGCTCTATGAGTGGCTA	55 °C	482	[31]
		CTCGAATGCCTGGCGTGTTT			
	*dfrA*	CCTTGGCACTTACCAAATG	50 °C	374	[32]
		CTGAAGATTCGACTTCCC			
	*sul1*	TGGTGACGGTGTTCGGCATTC	62 °C	789	[33]
		GCGAGGGTTTCCGAGAAGGTG			
	*sul2*	CGGCATCGTCAACATAACC	50 °C	722	[34]
		GTGTGCGGATGAAGTCAG			
	*sul3*	GAGCAAGATTTTTGGAATCG	51 °C	792	[35]
		CATCTGCAGCTAACCTAGGGCTTTGGA			
	*cmlA*	TGTCATTTACGGCATACTCG	55 °C	455	[28]
		ATCAGGCATCCCATTCCCAT			
	*floR*	CACGTTGAGCCTCTATAT		868	
		ATGCAGAAGTAGAACGCG			
Integrases	*int1*	GGGTCAAGGATCTGGATTTCG	62 °C	483	[33]
		GGGTCAAGGATCTGGATTTCG			
	*int2*	CACGGATATGCGACAAAAAGGT		788	
		GTAGCAAACGAGTGACGAAATG			
Virulence Factors	*fimA*	GTTGTTCTGTCGGCTCTGTC	55 °C	447	[36]
		ATGGTGTTGGTTCCGTTATTC			
	*hylA*	AACAAGGATAAGCACTGTTCTGGCT	63 °C	1177	[37]
		ACCATATAAGCGGTCATTCCCGTCA			
	*papC*	GACGGCTGTACTGCAGGGTGTGGGG		328	
		ATATCCTTTCTGCAGGGATGCAATA			
	*aer*	TACCGGATTGTCATATGCAGACCGT		602	
		AATATCTTCCTCCAGTCCGGAGAAG			
	*cnf1*	AAGATGGAGTTTCCTATGCAGGAG		498	
		CATTCAGAGTCCTGCCCTCATTATT			
Phylogroups	*chuA*	GACGAACCAACGGTCAGGAT	55 °C	279	[19]
		TGCCGCCAGTACCAAAGACA			
	*yjaA*	TGAAGTGTCAGGAGACGCTG		211	
		ATGGAGAATGCGTTCCTCAAC			
	*tspE4.C2*	GAGTAATGTCGGGGCATTCA		152	
		CGCGCCAACAAAGTATTACG			

**Table 2 vetsci-12-00675-t002:** Characterization of the twenty *E*. *coli* chosen for whole genome sequence analysis.

Isolate	Phenotype	β-Lactamase Genes	Non-β-Lactamase Genes	Chromosomal Mutations	MLST Type	O-Serotype	Plasmid Replicons (Reference Plasmid)
JR25	AMP-TET	*blaTEM*-1B	*tetA*, *acrF*, *mdtM*, *terD*, *terW*, *terZ*	*glpT*_E448K	ST155	O5:H11	IncFIA, IncFIB, IncFII(pHN7A8), IncY
JR26	Susceptible	-	*acrF*, *mdtM*, *ermD*, *ermE*	*cyaA_*S352T, *glpT*_E448K	ST5273-1LV	O51:H45	IncFIB, IncY, IncFII (29), IncFII(pSFO)
JR29	AMP-C-CIP-NAL-SME-TET-TRI	*blaTEM*-1B	*catA1*, *cmlA1*, *aph(6)-Id*, *aph(3″)-Ib*, *aadA1*, *aadA2b*, *tetA*, *dfrA1*, *sul1*, *sul2*, *sul3*, *acrF*, *mdtM*, *ermE*, *qacE*, *qacL*, *terD*, *terW*, *terZ*	*gyrA* p.S83L, *glpT*_E448K	ST4980	O88:H7	IncFIB, IncHI2, IncHI2A, IncQ1
JR30	AMP-AML- CAZ-CEP-CTX-PIT-C-CIP-TET-TRI	*blaTEM*-30, *blaTEM*-1B, *blaTEM-207*, *blaOXA-*10	*floR*, *cmlA5*, *qnrS1*, *tetA*, *dfrA14*, *aadA1*, *ARR-2*, *acrF*, *mdtM*	*glpT*_E448K	ST994	O174:H7	IncFIA, IncFIB, IncY, IncFIC(FII), IncX1
JR32	AMP-CAZ-CEP-CTX-TET	*blaTEM*-126, *blaTEM*-106, *blaTEM*-1B, *blaTEM*-135	*tetA*, *acrF*, *mdtM*, *merR*	*glpT*_E448K	ST345	O8:H21	IncFIB, IncFII
JR34	AMP	*blaTEM*-1B	*acrF*, *mdtM*, *ermE*	*cyaA*_S352T, *glpT*_E448K	ST3107	O69:H38	IncFIB, IncFIC(FII), IncFII
JR35	AMP	*blaTEM*-1A	*acrF*, *mdtM*, *ermE*	*glpT*_E448K	ST201	O8:H19	IncFIB, IncFII
JR37	AMP	*blaTEM*-1A	*acrF*, *mdtM*, *ermE*	*glpT*_E448K	ST201	O8:H19	IncFIB, IncFII
JR39	Susceptible	-	*acrF*, *mdtM*	*glpT*_E448K, *uhpT*_E350Q	ST1056	O174:H28	
JR42	AMP-CAZ-CEP-CTX-CIP-NAL	*blaCTX-M*-1	*acrF*, *mdtM*, *ermE*	*gyrA* p.S83L, *cyaA*_S352T, *glpT*_E448K	ST93	O5:H10	IncFIB, IncFIC(FII), IncI1-I(Alpha)
JR43	AMP-C-CIP-NAL-CN-SME-TOB-TET	*blaTEM*-1A	*catA1*, *sat2*, *aadA9*, *aadA13*, *aadA1*, *mph(B)*, *aac(3)-IIa*, *tetB*, *sul1*, *acrF*, *mdtM*, *ermE*, *qacE*	*gyrA* p.S83L, *glpT*_E448K	ST665	NT:H4	IncFIB, IncFIC(FII)
JR46	Susceptible.		*acrF*, *mdtM*, *ermE*	*glpT*_E448K	ST602	NT:H21	IncFIB, IncFIC(FII), IncX1, Col(pHAD28)
JR48	AMP-CIP-NAL-SME-TRI	*blaTEM*-1C	*aadA5*, *dfrA17*, *sul2*, *acrF*, *mdtM*, *ermE*	*gyrA* p.S83L, *gyrA* p.D87N, *parC* p.E84G, *parC* p.S80I, *glpT*_E448K	ST2973	O93:H16	IncFIB, IncY, IncFII, IncFIB(pLF82-PhagePlasmid)
JR50	CIP-NAL	-	*acrF*, *mdtM*, *ermE*	*gyrA* p.S83L	ST2973	O176:H12	IncFIB, IncFII, p0111
JR52	AMP-TET	*blaTEM*-1A	*tetA*, *acrF*, *mdtM*, *ermD*, *ermE*, *merC*, *merP*, *merR*, *merT*	*cyaA*_S352T, *glpT*_E448K, *nfsA*_Q44STOP	ST117	O114:H4	IncFIB, IncFIC(FII), Col(MG828), Col156
JR53	AMP-TET	*blaTEM*-1C	*tetA*, *acrF*, *mdtM*, *ermE*, *pcoA*, *pcoB*, *pcoC*, *pcoD*, *pcoE*, *pcoR*, *pcoS*, *silA*, *silB*, *silC*, *silE*, *silF*, *silP*, *silS*	*glpT*_E448K	ST58	O29:H34	IncFIB, IncFII(pRSB107)
JR54	AMP-TET	*blaTEM*-1C	*tetA*, *acrF*, *mdtM*, *ermE*, *pcoA*, *pcoB*, *pcoC*, *pcoD*, *pcoE*, *pcoR*, *pcoS*, *silA*, *silB*, *silC*, *silE*, *silF*, *silP*, *silS*	*glpT*_E448K	ST58	O29:H34	IncFIB, IncFII(pRSB107)
JR56	AMP-CIP-TET	*blaTEM*-1B	*qnrS1*, *tetA*, *acrF*, *mdtM*, *ermE*, *terD*, *terW*, *terZ*	*glpT*_E448K	ST155	O8:H51	IncFII(29), IncX1
JR58	AMP-TET	*blaTEM*-1B	*aph(6)-Id*, *aph(3″)-Ib*, *tetA*, *acrF*, *mdtM*, *ermE*, *terB*, *terC*, *terD*, *terE*	*cyaA*_S352T, *glpT*_E448K	ST3107	O69:H38	IncFIB, IncFIC(FII), p0111
JR60	Susceptible	-	*acrF*, *mdtM*, *ermE*	*glpT*_E448K	ST1611	O125ab:H19	IncFIA, IncFIB, IncFIC(FII)

AML—amoxicillin; AMP—ampicillin; CEP—cephalexin; C—chloramphenicol; CAZ—ceftazidime; CIP—ciprofloxacin; CN—gentamicin; CTX—cefotaxime; NAL—nalidixic acid; PIT—piperacillin-tazobactam; SME—sulfamethoxazole; TET—tetracycline; TOB—tobramycin; TRI—trimethoprim; (-)—no genes were detected.

**Table 3 vetsci-12-00675-t003:** Characterization of the remaining 16 *E*. *coli* isolated from sternal bursitis in chickens.

Isolate	Antibiotic Resistance		Virulence Factors	Integrons	Phylogroup
	Phenotype	Genotype			
JR27	Susceptible	-	*fimA*	-	B1
JR28	Susceptible	-	*fimA*, *aer*	-	B1
JR31	Susceptible	-	*fimA*	-	A
JR33	Susceptible	-	*fimA*, *aer*	-	B1
JR36	Susceptible	-	*fimA*	-	B1
JR38	Susceptible	-	*fimA*	-	A
JR40	Susceptible	-	*fimA*, *aer*	-	B1
JR41	Susceptible	-	*fimA*, *aer*	-	B1
JR44	Susceptible	-	*fimA*	-	B1
JR45	Susceptible	-	*fimA*, *aer*	-	D
JR47	Susceptible	-	*fimA*, *aer*	-	B1
JR49	Susceptible	-	*fimA*	-	B1
JR51	CIP	-	*fimA*, *aer*	-	D
JR55	Susceptible	-	*fimA*, *aer*	-	B1
JR57	TET	*tetB*	*fimA*, *aer*	-	B1
JR59	Susceptible	-	*fimA*, *aer*	-	B1

- no genes were detected.

## Data Availability

The data and findings generated in this study are available within the article and its Appendix A. For additional information, interested readers may contact the corresponding author.

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
