# Peer review of "Genomic Analysis of Antibiotic Resistance and Virulence Profiles in *Escherichia coli* Linked to Sternal Bursitis in Chickens: A One Health Perspective"

_vetsci, 2025, doi:10.3390/vetsci12070675_

Round 1

Reviewer 1 Report

Comments and Suggestions for Authors

This study investigated Escherichia coli isolates obtained from sternal bursitis lesions in broiler chickens. The isolates were characterized through whole-genome sequencing and antimicrobial resistance profiling.

  1. A total of 20 out of the 36 E. coli isolates were selected for whole-genome sequencing. However, the criteria used for selecting these isolates should be clearly stated
  2. For the remaining 16 E. coli isolates, the presence of antimicrobial resistance genes and virulence-associated genes was assessed using PCR. To enhance the clarity and methodological transparency of the study, the specific primer sequences referenced should be provided.
  3. Throughout the manuscript, please ensure that all bacterial species names and gene names are italicized.

Author Response

This study investigated Escherichia coli isolates obtained from sternal bursitis lesions in broiler chickens. The isolates were characterized through whole-genome sequencing and antimicrobial resistance profiling.

  1. A total of 20 out of the 36  coliisolates were selected for WGS However, the criteria used for selecting these isolates should be clearly stated.

Thank you for your valuable comment. We agree that the selection criteria for whole-genome sequencing (WGS) should be clearly explained. Due to limitations in sequencing capacity within the scope of our collaboration, only 20 of the 36 E. coli isolates could be submitted for WGS. To maximize the relevance and diversity of the genomic data obtained, we selected isolates based primarily on their phenotypic antimicrobial resistance profiles. Specifically, we prioritized isolates showing multidrug resistance or distinct resistance patterns, aiming to capture a representative and informative subset of our collection. We have now clarified this point in the revised manuscript [see section 2.4.].

  1. For the remaining 16  coliisolates, the presence of antimicrobial resistance genes and virulence-associated genes was assessed using PCR. To enhance the clarity and methodological transparency of the study, the specific primer sequences referenced should be provided.

Thank you for your valuable suggestion. In response, we have included a detailed table (Table 1) listing all specific primer sequences used for PCR amplification of antimicrobial resistance genes, virulence factors, integrase genes, and phylogenetic markers in our Escherichia coli isolates. This addition improves the methodological transparency and reproducibility of our study.

  1. Throughout the manuscript, please ensure that all bacterial species names and gene names are italicized.

Thank you for your observation. We have carefully reviewed the manuscript and confirm that all bacterial species names are now correctly italicized throughout the text. Gene names have also been checked and formatted according to the journal’s guidelines.

Reviewer 2 Report

Comments and Suggestions for Authors

The manuscript addresses a highly important topic — antimicrobial resistance — within a unique and specific context involving virulence factors and bursitis in chickens. The manuscript is well-written and highly relevant; however, there are several aspects that require attention:

  1. In the Introduction, it would be beneficial to include a few sentences on the economic significance of domestic chickens — particularly broiler chickens — and highlight the economic losses associated with sternal bursitis (primarily due to carcass condemnation).
  2. Line 103: Please clarify the time interval during which sampling took place. Why were 40 samples collected specifically? What was the prevalence of lesions in the affected chickens?
  3. Line 180: It is not customary to merge the Results and Discussion sections. This should be revised accordingly. First, the authors should present their findings clearly in a separate Results section and then interpret and compare these findings with those reported in other studies in a distinct Discussion section.

Author Response

The manuscript addresses a highly important topic — antimicrobial resistance — within a unique and specific context involving virulence factors and bursitis in chickens. The manuscript is well-written and highly relevant; however, there are several aspects that require attention:

  1. In the Introduction, it would be beneficial to include a few sentences on the economic significance of domestic chickens — particularly broiler chickens — and highlight the economic losses associated with sternal bursitis (primarily due to carcass condemnation).

Thank you for your valuable suggestion. I have revised the Introduction to include a brief section on the economic importance of domestic chickens, particularly broilers, and emphasized the economic losses associated with sternal bursitis, especially due to carcass condemnation.

  1. Line 103: Please clarify the time interval during which sampling took place. Why were 40 samples collected specifically? What was the prevalence of lesions in the affected chickens?

Thank you for your observation. We have clarified the sampling period in the revised manuscript (section 2.1.). Sampling was conducted between November and December 2021. The 40 samples were collected based on the number of affected carcasses provided by a veterinary medicine collaborator during routine post-mortem inspections. These represented all available birds with visible sternal lesions at the time. Unfortunately, data on the total number of chickens inspected were not recorded, preventing us from calculating the prevalence of lesions. This limitation has been acknowledged in the revised manuscript.

  1. Line 180: It is not customary to merge the Results and Discussion sections. This should be revised accordingly. First, the authors should present their findings clearly in a separate Results section and then interpret and compare these findings with those reported in other studies in a distinct Discussion section.

Thank you for your comment. While we acknowledge that separating the Results and Discussion sections is a common structure, many peer-reviewed articles in our field also adopt a combined “Results and Discussion” format, particularly when the interpretation of findings is closely tied to their presentation. As none of the other reviewers raised concerns regarding this structure, and considering the nature and flow of our data, we believe that the merged format provides a clearer and more coherent narrative in this case. We have therefore opted to retain the combined section.

Reviewer 3 Report

Comments and Suggestions for Authors

An interesting study; however, the authors should incorporate some minor revisions.

Line 104 : Is it known if the samples were taken from carcasses from the same farm? And how many farms were analyzed?

Line 198 From this line onward, the font size is smaller compared to the rest of the text.

Results and disscusin: The authors do not explain why they used such antibiotics; they only mention in line 224 that differences in antimicrobial resistance may result from differences in legislation and antibiotic use policies. For readers unfamiliar with food and veterinary law, it would be useful to highlight the latest requirements and the One Health approach, as the EU Regulation 2019/6 and its Implementing Regulation 2022/1255 establish strict bans on certain antibiotic groups to preserve human‑critical antimicrobials.

According to the current approach, some antibiotics are banned for use in animals. In the USA, this group includes fluoroquinolones, while under EU regulations, the use of aminopenicillins with beta-lactamase inhibitors, carbapenems, penems, monobactams, etc., is prohibited in food-producing animals.

This is laid down in the EU Veterinary Medicinal Products Regulation (EU) 2019/6, effective from 28 January 2022, which fully bans the use of certain antimicrobial classes reserved for human medicine, and was further reinforced by Commission Implementing Regulation (EU) 2022/1255, specifying the list of prohibited agents in food‑producing species . This study may suggest that the restrictive European legislation on antibiotic use reduces the frequency of antimicrobial resistance, and this is worth highlighting.

Author Response

An interesting study; however, the authors should incorporate some minor revisions.

Line 104 : Is it known if the samples were taken from carcasses from the same farm? And how many farms were analyzed?

Thank you for your question. We have clarified in the revised manuscript (Line 104) that all samples were collected from carcasses originating from the same farm. Therefore, only one farm was analyzed in this study.

Line 198 From this line onward, the font size is smaller compared to the rest of the text.

Thank you for pointing this out. We have carefully reviewed the formatting and can confirm that the font size remains consistent throughout the manuscript

Results and disscusin: The authors do not explain why they used such antibiotics; they only mention in line 224 that differences in antimicrobial resistance may result from differences in legislation and antibiotic use policies. For readers unfamiliar with food and veterinary law, it would be useful to highlight the latest requirements and the One Health approach, as the EU Regulation 2019/6 and its Implementing Regulation 2022/1255 establish strict bans on certain antibiotic groups to preserve human‑critical antimicrobials. According to the current approach, some antibiotics are banned for use in animals. In the USA, this group includes fluoroquinolones, while under EU regulations, the use of aminopenicillins with beta-lactamase inhibitors, carbapenems, penems, monobactams, etc., is prohibited in food-producing animals. This is laid down in the EU Veterinary Medicinal Products Regulation (EU) 2019/6, effective from 28 January 2022, which fully bans the use of certain antimicrobial classes reserved for human medicine, and was further reinforced by Commission Implementing Regulation (EU) 2022/1255, specifying the list of prohibited agents in food‑producing species . This study may suggest that the restrictive European legislation on antibiotic use reduces the frequency of antimicrobial resistance, and this is worth highlighting.

We thank the reviewer for the valuable suggestion regarding the clarification of antibiotic selection and the inclusion of recent regulatory context. In response, we have expanded the discussion to explicitly address the relevance of the antibiotics tested in relation to current European Union legislation, specifically Regulation (EU) 2019/6 and Commission Implementing Regulation (EU) 2022/1255, which restrict the use of critically important antimicrobials in food-producing animals. We have also emphasized the importance of the One Health approach in preserving the efficacy of last-resort antibiotics. These additions aim to provide a clearer understanding of how regional antibiotic use policies influence resistance patterns observed in our study.

Round 2

Reviewer 2 Report

Comments and Suggestions for Authors

The authors have corrected my comments and the quality of the manuscript has improved significantly.